# Direct-laser writing for subnanometer focusing and single-molecule imaging

Simao Coelho[1,2,3,4 ✉], Jongho Baek [1,2,7], James Walsh[1,2], J. Justin Gooding [5,6 ✉] & Katharina Gaus [1,2,8]

Two-photon direct laser writing is an additive fabrication process that utilizes two-photon absorption of tightly focused femtosecond laser pulses to implement spatially controlled polymerization of a liquid-phase photoresist. Two-photon direct laser writing is capable of nanofabricating arbitrary three-dimensional structures with nanometer accuracy. Here, we explore direct laser writing for high-resolution optical microscopy by fabricating unique 3D optical fiducials for single-molecule tracking and 3D single-molecule localization microscopy. By having control over the position and three-dimensional architecture of the fiducials, we improve axial discrimination and demonstrate isotropic subnanometer 3D focusing (<0.8 nm) over tens of micrometers using a standard inverted microscope. We perform 3D single-molecule acquisitions over cellular volumes, unsupervised data acquisition and live-cell single-particle tracking with nanometer accuracy.

[1] EMBL Australia Node in Single Molecule Science, School of Medical Sciences, University of New South Wales, Sydney, NSW, Australia. [2] ARC Centre of Excellence in Advanced Molecular Imaging, University of New South Wales, Sydney, NSW, Australia. [3] Structural Biology Program, Memorial Sloan Kettering Cancer Center, New York, NY, USA. [4] Instituto Gulbenkian de Ciência, 2780-156, Oeiras, Portugal. [5] School of Chemistry and Australian Centre of NanoMedicine, University of New South Wales, Sydney, NSW, Australia. [6] ARC Centre of Excellence in Convergent Bio-Nano Science and Technology, University of New South Wales, Sydney, NSW, Australia. [7] Present address: NetTargets, National Nanofab Center, KAIST, Daejeon, Republic of Korea. [8] Deceased: Katharina Gaus. ✉email: s.pereiracoelho@unsw.edu.au; justin.gooding@unsw.edu.au

Single-molecule microscopy has greatly expanded our knowledge of structural organizations, functional conformations and dynamics of protein complexes within cellular environments. Remarkable progress has been made in single-molecule imaging methods to improve spatial resolution[1–3], penetration depth[4–6], and live-cell imaging capabilities[7,8].

Single-molecule imaging methods with high spatial resolutions all employ a combination of axial focus-locking (e.g., infrared laser in total internal reflection mode) and lateral correction methods (e.g., fluorescent markers). Real-time 3D focus-locking performed with high-accuracy (~1 nm) maximizes the photon collection from individual fluorescent events and has shown a > 10-fold improvement in the localization precision over standard methods without active stabilization[9]. Inaccurate or slow active corrections result in drift degrading the localization precision and considerably inferior in situ resolution, even after post-analysis treatment such as filtering or grouping[10].

Real-time focus locks with nanometer precision have been applied to in vitro samples by combining optical trapping and optimizing on the $x/y$ position and width ($z$) of single emitters in isolation[9]. Recent developments, which are compatible with cellular imaging, rely on the deposition of fiducials at random[10–12] or the transmitted profile of the specimen itself in brightfield images[13]. However, when imaging at depths >5 μm from the coverslide, none of these methods have been shown to significantly improve on commercially available axial focus locks that typically have ~20 nm precision (i.e., Nikon Perfect Focus) and provide limited—if any—lateral compensation. Thus, despite the promise of fluorescence imaging technologies capable of high localization precision with increased penetration depth, their general adaption to 3D imaging remains limited. Hence, a focusing method that has live-cell compatibility, absolute 3D positioning across micrometer ranges and an accuracy better than the photon-limited localization precision of any previously reported single-molecule method is highly desirable.

Here, we address these focusing limitations by using two-photon direct-laser writing to control the position and three-dimensional architecture of the fiducials in the sample. In doing so, we reproducibly achieve real-time subnanometer focusing in all dimensions across cellular volumes. We demonstrate the versatility of our approach by acquiring 3D single-molecule acquisitions and live-cell single-particle tracking with nanometer accuracy

## Results

**Two-photon direct-laser writing for subnanometer focusing**. Here, we developed custom 3D fiducial markers via two-photon direct-laser writing[14,15]. Our 3D fiducial architecture enables isotropic focus locking directly on to the imaging plane and absolute positioning with 0.8 nm. We performed 3D nanoscale fabrication by solidifying a liquid photoresist under two-photon illumination[14–16] (Fig. 1). The optical fiducials were fabricated with a commercially available femtosecond laser lithography system (Nanoscribe GmbH, Germany). Computer-aided design software allows for straightforward in situ 3D printing with significant geometric versatility. By precisely scanning the laser focal volume along a designated path, fine 3D structures can be printed with sub-micrometer features (Supplementary Fig. 1). 3D samples have an accuracy between 160 and 200 nm, which is routinely achieved using the Nanoscribe. Careful scanning allows for structures to be produced with a surface roughness of 13.0 nm[15]. Previous studies have shown that by manipulating materials and fabrication techniques, finer structures with features of ~30 nm can be produced successfully[17]. This unique property has been exploited for manipulating cells in scaffolds[18], the

generation of ultracompact compound lenses[15] and mechanical metamaterials[19].

The transparent photoresist makes direct-laser writing suitable to create optical structures[15] which can be recorded using transmitted light in brightfield. The diffraction pattern provides information on both the $x$–$y$ and $z$ position. The interference pattern from the photopolymerized fiducials were recorded on a dedicated CMOS camera with a high frame rate (~400 Hz) and large field-of-view. The structures were captured simultaneously with the single-molecule fluorescence of the sample. The wavelength of the transmitted light was selected to avoid cross-excitation or contamination of the weak fluorescent signal.

To achieve real-time subnanometer focusing, we use active stabilization and obtain a stabilization of 0.4 nm (standard deviation) in the lateral ($x$–$y$) and 0.7 nm in the axial ($z$) direction, over hours and days (Fig. 2, Supplementary Fig. 2, Supplementary Table 1, and Methods). The stability of the sample is determined as the standard deviation of the position of the fiducial recorded as a function of time. The sample stability is recorded independently using a second structure as a reference. The reference structure is an out-of-loop fiducial whose deviations are calculated independently from the focus-locking routine. Focus locking can also be monitored by recording the difference between the current and original position ($x/y/z$) at the start of the experiment. We use GPU accelerated calculations to provide real-time stage corrections with a speed of 20 Hz. The accuracy of the focus lock shown here was better than the localization precision of any previously reported single-molecule method to date[1–11,13,20–24].

The flexibility of the fabrication process allows for fiducial designs with varying 3D distributions, arrays of fiducials with variable spacing and in situ nanoprinting at predefined locations. By producing fiducial arrays with different geometries, dimensions and distributions, we created prefabricated coverglass capable of focus locking, via their brightfield profile, across a large range of imaging volumes (Fig. 2e, f and Supplementary Videos 1–4). Furthermore, by modifying the spherical cap of fiducials in contact with the coverslide, the contrast of the diffraction pattern was enhanced, and axial stabilization improved to 0.65 nm (Supplementary Fig. 3). This opens the possibility for non-symmetrical designs and structures, which could further enhance performance and imaging range.

Direct-laser writing removed the need for multiple fiducials per imaging plane, as residual errors (which originate from surface-linkage errors) were eliminated and post-acquisition corrections were not required. The 3D architecture allows for subnanometer focus locking far beyond the coverslip and without the need of a complex optical setup.

**Custom fiducials for single-molecule imaging and live-cell single-particle tracking**. To assess the focusing capabilities for single-molecule imaging, we first demonstrated our approach by imaging microtubules in COS-7 cells (Fig. 3). The fiducials used were printed spheres with a 20% spherical cap enclosed in a nano-fabricated box to minimize contamination from cellular debris and to protect them from cell overgrowth (Fig. 3a, b).

Direct-laser writing produces optical fiducials that are part of the coverslide and are fully biocompatible. This allows for cells to be grown directly onto the imaging slide and thus does not affect sample preparation. COS-7 cells were grown directly on the imaging slide containing the nano-fabricated fiducials. We performed 3D direct stochastic optical reconstruction microscopy (dSTORM) under total internal reflection (TIRF) by introducing a cylindrical lens[24]. Figure 3c shows the distribution of the microtubules and three-dimensional false-color distribution. An

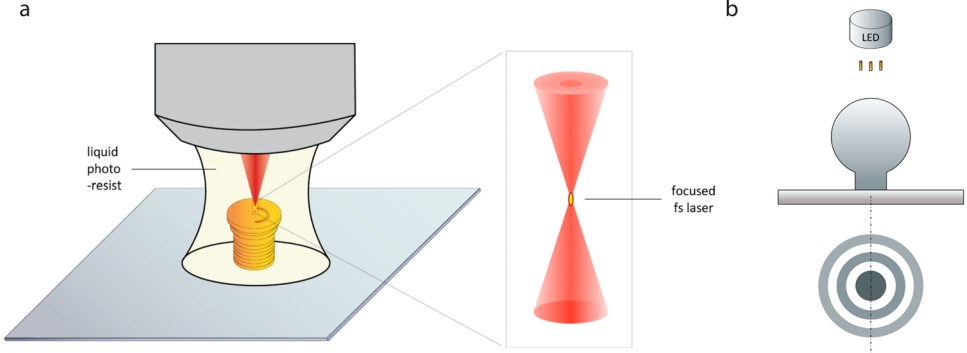

**Fig. 1 Illustration demonstrating fiducial fabrication and 3D focus locking. a** Two-photon direct-laser writing. A NIR femtosecond laser is used to trigger a chemical reaction that causes polymerization of a photosensitive material. The nonlinear excitation solidifies the photo resin at the focal point, leaving other regions unaffected. **b** 3D focus locking. Under brightfield illumination the fiducials create an interference pattern (below) which is recorded on a separate camera with high frame rate. Changes in the diffraction pattern are used to monitor the movement experienced by the sample.

estimated value of 50 nm (full width at half maxima) was obtained for the diameter of a microtubule, as expected due to the additional size of the primary and secondary antibodies[25] (Fig. 3d and Supplementary Fig. 4).

As the nanoprinting is performed with high precision, the interference pattern for a specific fiducial design was reproduced systematically across samples. Our approach incorporated an automatic search and recognition of fiducials printed on the glass slide and matched the registered diffraction pattern to an axial offset from the surface (Supplementary Fig. 5). By using prior knowledge of the shape and distribution of the fiducials located throughout the imaging slide, we developed an automated acquisition platform to perform autonomous unsupervised data acquisition (Fig. 3c).

To assess the three-dimensional localization precision, clusters of localization points for individual Alexa-Plus 647 molecules were selected and aligned by their centers of mass and fit to a Gaussian distribution, yielding standard deviations of ~6 nm along $x$ and $y$ and ~13 nm in the $z$ direction (Fig. 3e–g), an improvement of ~2-fold over standard 3D imaging[24].

To evaluate the live-cell imaging capabilities, we performed single-particle tracking (SPT) in living HeLa cells (Fig. 3h–l). In live-cell SPT, fiducials are used to differentiate between the movement of the cell, the trajectory of the particle and sample drift. The cells were acquired using highly inclined and laminated optical sheet (HILO) illumination[6] to capture three-dimensional images (Fig. 3h–j). The height $z$ from the coverslip was made by offsetting the axial focus lock on the fiducial. We demonstrate live-cell SPT by recording the diffusion of individual CD47 molecules (Fig. 3k, l) in the plasma membrane.

To compare the performance our of approach to commonly used focus-locking mechanisms, we performed experimental evaluations of: (1) a commercially available focus lock (Nikon Perfect Focus System) and (2) image correlation correction using the contrast from cellular structures (Supplementary Fig. 6). We found our method improves on commercially focus locks and image correlation by ~20-fold. To validate our approach, we deposited fluorescent fiducials on top on cells at a depth of 5 μm (Supplementary Fig. 7). Over ~3 h, we registered <2 nm variation in position. This is consistent with the photon-limited localization precision recorded, thus demonstrating that the fluorescence remained focused and stable.

**3D super-resolution imaging of over tens of micrometers.** Next, we demonstrated the versatility of our approach by acquiring 3D super-resolution images over entire cellular volumes using single-objective selective-plane illumination microscopy (soSPIM)[4]. In

soSPIM, an excitation plane is axially translated throughout the sample by reflecting a light-sheet off a 45° mirror embedded within the sample chamber. Fluorescence from each individual excitation light-sheet plane was collected using the same high numerical aperture objective (Fig. 4a).

We were able to nanoprint optical fiducials directly within individual wells of the imaging chambers (Fig. 4b) and performed light-sheet imaging using standard soSPIM (Fig. 4c) and soSPIM-dSTORM over multiple axial sections (Fig. 4d). Axial planes to cover a depth of 20 μm were recorded by focus locking the excitation light-sheet over multiple z-sections, while maintaining the lateral position. Since the 45° mirror associates lateral movement (i.e., $\Delta x$) with translation of the excitation plane in $z$, 3D stability is particularly important for long imaging periods (e.g., dSTORM). The 3D soSPIM single-molecule localization data produced did not require further post-acquisition corrections. Thus, we overcome a current obstacle in 3D imaging, which is the lack of suitable fiducials that can be recorded at depth. By successfully 3D focus locking across entire cellular volumes, we captured cross-sections of the distribution of CD47 along the membrane of HEK293 cells deposited within the imaging chambers (Fig. 4).

## Discussion

In summary, we present a method which provides isotropic 3D focusing and focus locking with subnanometer precision across tens of micrometers. Until now, fiducials for single-molecule imaging were deposited at random throughout the imaging slide, provided limited axial discrimination and do not show the 3D focusing possibilities presented here. We presented a solution for lateral and axial focusing at depth with a ~20-fold improvement over current focusing methods. Using prefabricated fiducial slides with advanced 3D single-molecule imaging techniques can thus help bridge the gap between high-resolution and penetration depth, while retaining live-cell compatibility.

In principle, the focusing capabilities can be further extended beyond cellular volumes, such as hundreds of microns, by tailoring the fiducial design (Supplementary Fig. 8), using a combination of fiducial markers in tandem and/or simultaneous multi-fiducial tracking. In contrast to traditional lithography techniques, two-photon direct-laser writing is particularly well suited to produce complex three-dimensional structures, such as helixes, which can span across millimeter ranges. Such geometries could potentially cover the entire penetration range of fluorescence optical microscopy.

Currently, producing highly accurate structures using direct-laser writing can be a slow, expensive process as fabrication is

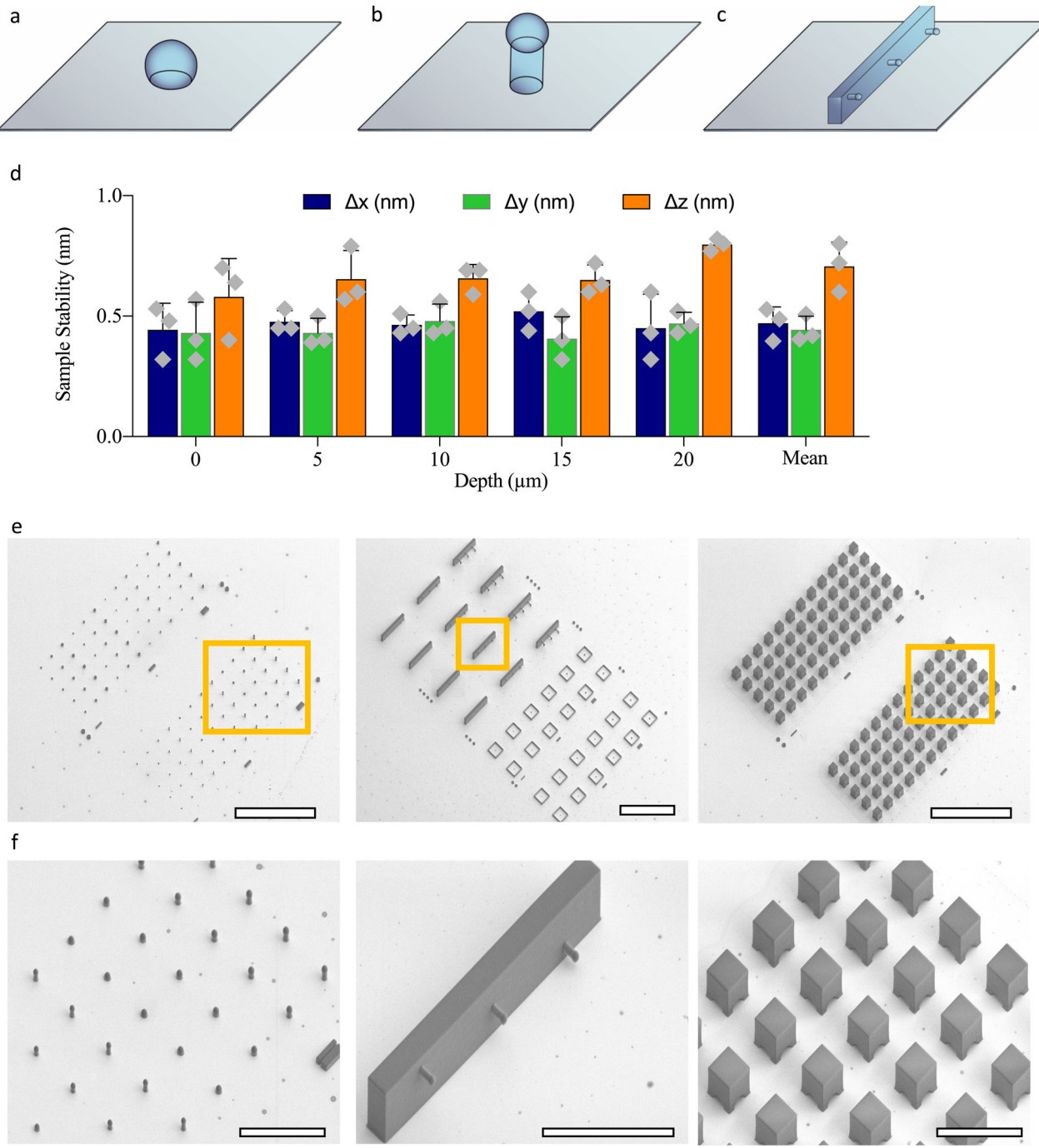

**Fig. 2 Implementation of two-photon direct-laser writing for 3D subnanometer focusing. a–c** Schematics of 3D nanoprinted fiducials. **a** A 3 μm diameter and a 20% spherical cap (**b**) fiducial (same shape as in **a**) on a 3 μm high pedestal and **c**, fiducials assembled in a 3D architecture. The design in (**c**) covers a vertical range of 20 μm. **d** Subnanometer real-time 3D focus locking. Focus-locking was performed from 0 to 20 μm in 1 μm increments. Each z-step was acquired for 1 h (*n* = 3), with a combined duration of 63 h. The residual movement (*Δx/Δy/Δz*) was determined using a duplicate out-of-loop structure acquired simultaneously. Data are presented as mean values of the residual movement and error bars correspond to the standard deviation across the experimental repeats (*n* = 3). The sample stability across all depths in all directions was less than 0.8 nm. **e, f** are examples of prefabricated fiducial arrays. **e** SEM images of multiple fiducial arrays (*n* = 15 arrays). The arrays on the right have spherical fiducials enclosed within a protective box (right). **f** Zoom-in of highlighted areas in (**e**). Scale bars = 200 μm in (**e**) and scale bars = 50 μm in (**f**).

performed layer-by-layer using two-photon lithography. We expect that advancements in fabrication procedures, including parallelization[26] and photopolymers, will make this more accessible to wider scientific community and even mass-producible[27]. It is also worth noting that incorporating high-accuracy focus locking can prove beneficial to improving optical techniques with high collection efficiency and penetration, such as lattice light sheet with adaptive optics[28] and/or high photon count DNA-PAINT approaches[29].

Due to its simple implementation and compatibility with standard microscopy hardware, our approach can be easily integrated in a wide range of techniques beyond single-molecule fluorescence. The fiducials can be used in imaging modalities which benefit from precise focus-locking (e.g., atomic force

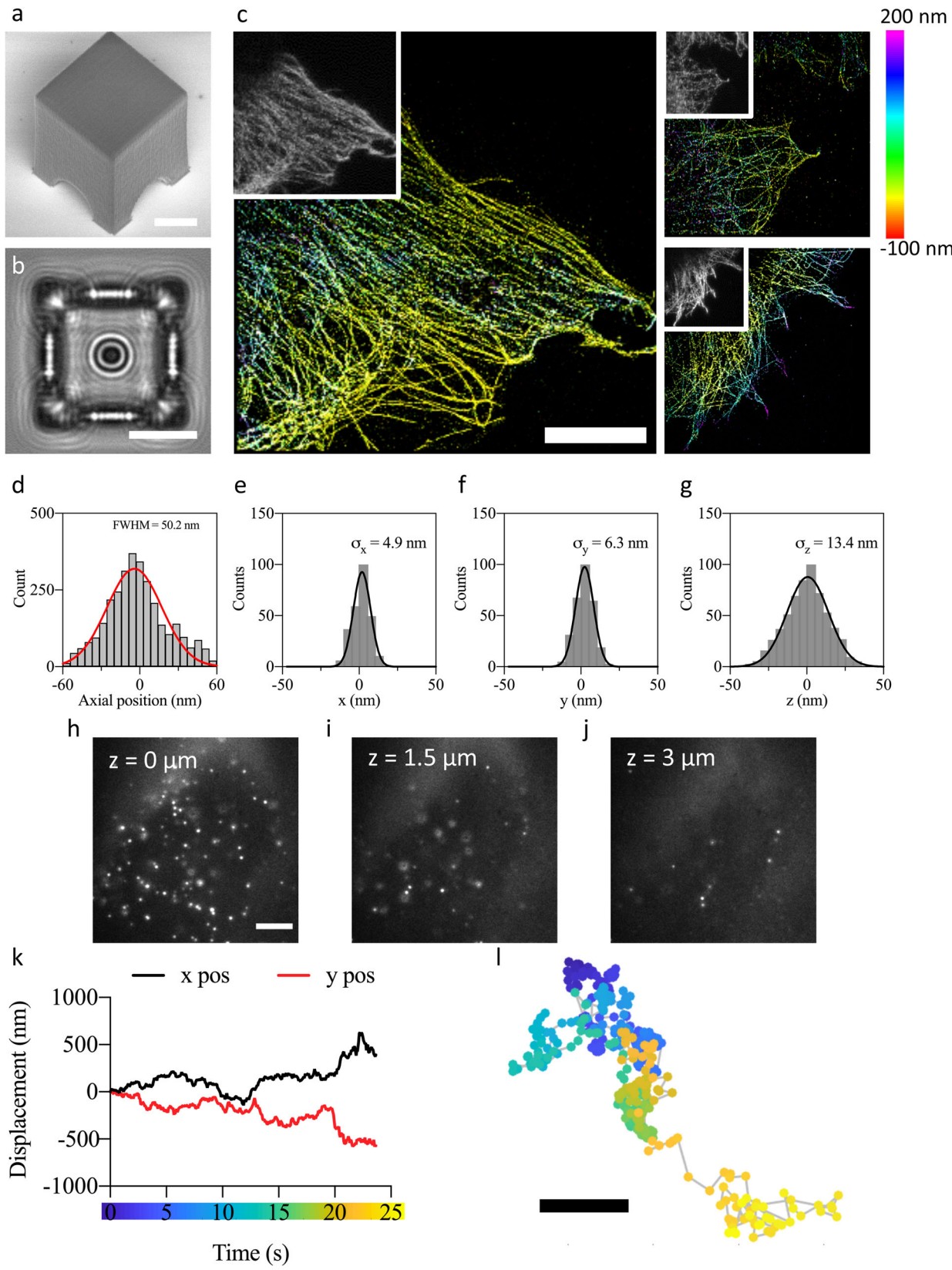

microscopy), have inherent optically sectioning or in nano/microfabrication techniques.

## Methods

**Direct-laser writing.** Light from a two-photon laser is used to trigger a chemical reaction that causes polymerization of a photosensitive material. The nonlinear excitation occurs at the focal point leaving other regions unaffected by the laser light. This phenomenon reduces solidification resolution to below the diffraction limit of the applied light. By selectively solidifying the photo resin, 3D geometries are produced layer-by-layer. Samples were fabricated by 3D direct-laser writing (Nanoscribe GmbH, Photonics Professional GT) using a commercially available photoresist (IP-Dip, Nanoscribe GmbH)[30]. The ultraviolet curable liquid resist was polymerized via two-photon absorption at a wavelength of 780 nm. The designed

**Fig. 3 Custom fiducials for 3D dSTORM imaging, unsupervised data acquisition and live-cell single-molecule tracking. a** SEM image of a spherical fiducial enclosed in a nano-fabricated protection box ($n = 60$ per array). **b** Transmitted diffraction profile of the fiducial inside the box. **c** 3D dSTORM images of the microtubule network in fixed COS-7 cells color-coded as indicated by the colored depth ($n = 12$). Cells were grown directly on the coverslide and acquired in an unsupervised manner. Inset shows the diffraction limited image. **d** Axial profile of a microtubule in (**c**) has a resolution of 50.2 nm. **e–g** Clusters of localization points for individual Alexa-Plus 647 molecules were selected and aligned by their centers of mass and fit to a Gaussian distribution. The standard deviation was 4.9 nm in the $x$-direction (**e**), 6.3 nm in the $y$-direction (**f**) and 13.4 nm in the $z$-direction (**g**). **h–l** Single-molecule tracking in living cells. Representative three-dimensional images images of CD47 molecules in live HeLa cells acquired using highly inclined and laminated optical sheet (HILO) microscopy. A time series was acquired at the surface (**h**), a depth of 1.5 μm (**i**), and 3 μm (**j**) ($n = 4$). **k–l** Details of a track. $x$ and $y$ displacement of a representative CD47 molecule (**k**) and 2D representation (**l**) using the color-coded time scale in (**k**). Scale bars = 10 μm in (**a–c**), 5 μm in (**h**) and 200 nm in (**l**).

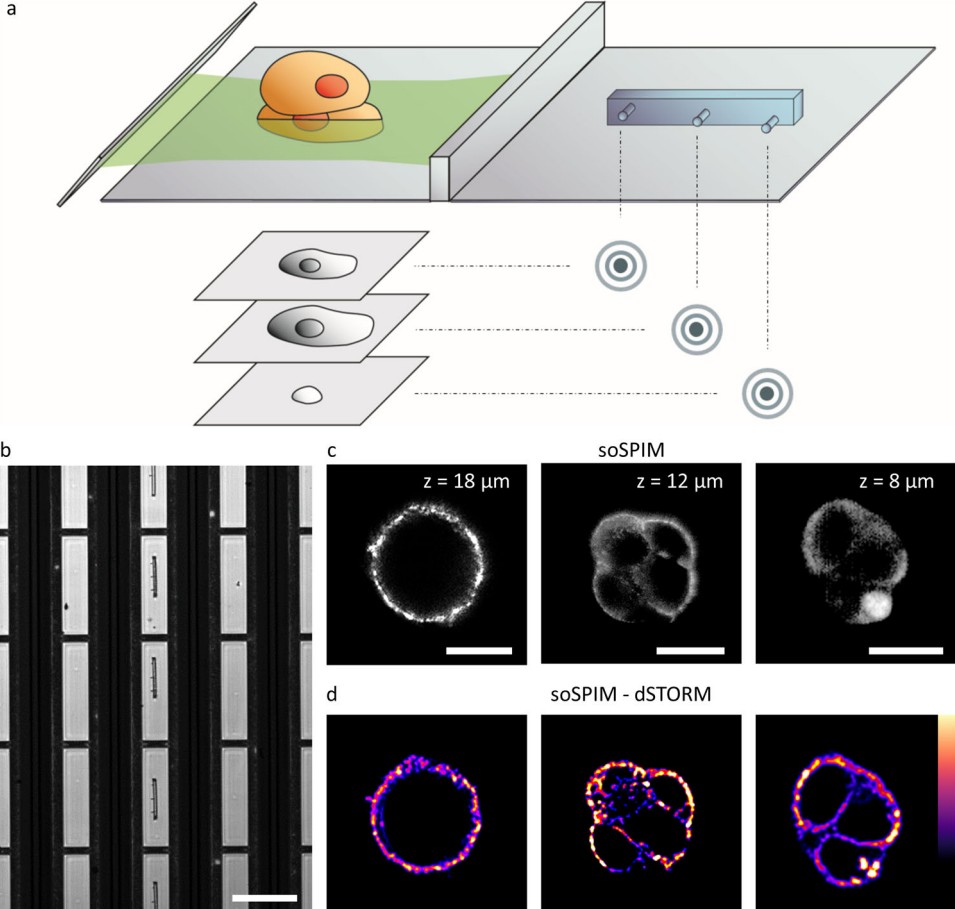

**Fig. 4 3D focus locking for super-resolution single-objective selective-plane illumination microscopy over cellular volumes. a** Laser light is reflected off a 45° mirror selectively illuminating a single plane within the sample. Fluorescence is captured via the same objective. Focus locking onto different axial planes is performed by using the 3D fiducial. **b** Brightfield image of 3D fiducial structures (shown schematically in Fig. 2c), taken with a x10 objective, located within individual wells of the soSPIM imaging chambers (central column) ($n = 7$ chambers). Scale bar = 200 μm. **c** soSPIM optical sections showing the distribution of CD47 labeled with Alexa-Plus 647 in HEK293 cells at the indicated depths ($n = 6$). **d** Corresponding dSTORM images of CD47 at the depths shown in (**c**) ($n = 6$). Scale bar = 5 μm.

structures were written layer-by-layer with 100 nm slicing distance between the individual layers. Computer-aided design of the fiducial structures were produced using Autodesk fusion 360 and exported as stereolithography (.STL) files.

**Cell culture, fixation, and immunostaining**. COS-7, HeLa and HEK293T cells were cultured in DMEM (Life Technologies, 11885-084) supplemented with 10% FBS and 1% penicillin–streptomycin. All cell fixation and immunostaining were done directly on coverslip/chambers with prefabricated photopolymerized fiducials.

For COS-7 cells, a coverslip was attached to an 8-well chamber (ibidi 80841) and washed with PBS. Approximately 10,000 cells were added per well and incubated for 72 h at 37 °C. Cells were fixed in 4% paraformaldehyde solution (PFA, Sigma), washed using a solution of PBS and incubated with 5% BSA/PBS for 1 h to avoid nonspecific binding. Cells were permeabilized with a 0.1% solution of

Triton X-100. Cells were incubated overnight at 4 °C in a 1:1000 dilution of anti-alpha-tubulin antibody (DM1A, Sigma-Aldrich), washed with PBS and incubated for 45 min with a secondary antibody labeled with Alexa Fluor Plus 647 (A32728, Life Technologies) using a 1:500 dilution.

For HeLa cells, a coverslip was attached to an 8-well chamber (ibidi 80841) and washed with DMEM. Approximately 10,000 cells were added per well and incubated for 24 h at 37 °C. The cells were labeled using a 1:50,000 dilution of CD47 (Invitrogen, B6512) conjugated to Alexa Fluor Plus 647 (A32728, Life Technologies).

HEK293T cells were deposited directly within the dish containing the microwells for the soSPIM imaging. First, DMEM media was added to the microwells by placing the chamber in a vacuum chamber for 30 min. Approximately 10,000 cells were deposited in the microwells by centrifugation at $50 \times g$ for 1 min and incubated at 37 °C overnight. Fixation was performed using 4% PFA. Cells were permeabilized with a 0.1% solution of Triton X-100, washed

with PBS and incubated with 5% BSA/PBS for 1 h. A 1:500 dilution of CD47 (Invitrogen, B6512) was added for 30 min, washed with PBS and incubated for 45 min with a secondary antibody labeled with Alexa Fluor Plus 647 (A32728, Life Technologies) using a 1:200 dilution in 5% BSA/PBS. The sample was washed using PBS, and then fixed using 4% PFA for 15 min.

**Unsupervised data acquisition**. To perform autonomous acquisitions, we used coverslips with arrays of fiducial structures. Fiducials were produced with a regular spacing which matched the field-of-view, thus ensuring at least one per area. The geometry and distribution of the fiducial array was added to the acquisition software which enables automatic recognition of fiducials within the array. The diffraction pattern for individual fiducials was also added to the software. The diffraction profile is a look-up table made up of the average radial profile as a function of depth. Direct-laser writing removes the large range of sizes commonly found in other fiducials (e.g., latex beads) and thus, allows for comparable axial look-up tables. Prior to determining a new fiducial's position, we typically use an average of 10–30 diffraction profiles with the same geometries.

Initially the surface was scanned in a *xy* raster pattern. The software can automatically identify a fiducial once it enters the field-of-view. Initially, the *xy* position of a fiducial is estimated, within ~1 pixel, by using cross-correlation. Fiducials are then identified by their spacing, predicated geometry and distribution throughout the slide. After a fiducial is identified their 3D position is determined via their individual look-up table.

At each fiducial, the surface was located by iteratively performing z-stacks. In each iteration, the axial range of the stack was reduced. Once the surface was found, an axial profile of the fiducial was acquired by performing a z-stack over a pre-determined axial range by stepping the nano-positioning stage in 5 nm increments. Variations in print design and real shape of the beads is not a problem since the interference pattern of the bead in question is measured before the experiment. The rotational orientation of the array (sample chamber) was determined by calculating the distance between fiducials distributed throughout the slide. For single-molecule imaging, cellular structures or regions-of-interest (ROI) were selected and the software automatically acquired images for each location sequentially.

The accuracy of the automatic surface detection routine is 35 nm (Supplementary Fig. 5) and was determined by recording the height difference between our autonomous focusing method and the position with maximum signal of individual molecules deposited onto the coverslide. Errors from the estimated position of the surface and the surface itself can arise from a variety of factors such as surface flatness. In practice, this error can be reduced by manually (re)focusing onto the fluorescence from the sample prior to imaging. The focus-locking routine, which maintains the sample stationary, is activated after the fiducial is found on the surface and the performance is not influenced by the variations of the automatic surface detection routine.

**3D focus locking**. Real-time focus locking was performed by repositioning the sample using the nano positioning stage at a frequency of 20 Hz. Applying an equal but opposite amplitude suppressed mechanical movement. Focusing onto a different z-position was done by offsetting the lock-in plane across the diffraction pattern[12]. To change the lateral position, an offset ($\partial x$, $\partial y$) is added to the lock-in position. The software automatically focus-locked onto the new location. For complex assemblies, such as the vertical wall, each individual sub-element covered an axial range. The structures were designed such that the axial profiles of each element overlap in z, but remain separate in x/y. The full focusing range is determined by the number of sub-elements and the axial range of the diffraction pattern.

Calculations were performed by using a camera-based particle tracking software implemented in LabVIEW[31]. We used real-time data processing in a graphics-processing unit (CUDA, NVIDIA GTX 1080). The lateral positions were determined by correlating linear intensity profiles with their mirror profiles[10,32]. The axial position was determined by comparing the radial intensity distribution to a pre-recorded look-up table. To enhance the signal from the diffraction patterns, we used a nonlinear dynamic fixed pattern noise correction[31]. This corrects for the nonlinear pixel response, read-out noise and inhomogeneous illumination. CMOS technology allows for an increased frame rate by selectively recording a ROI. The diffraction patterns from the fiducial structures were captured at ~400 Hz during the single-molecule acquisitions (10–15 Hz).

To ensure high contrast in brightfield, we selected high N.A. objectives. These were previously inspected by the manufacture using wavefront aberration technologies to ensure the lowest possible asymmetric aberration and superior optical performance required for super-resolution imaging.

The diameter of the fiducial can influence performance. To determine the best fiducial dimension for a given objective/magnification, we empirically evaluate each imaging objective using an array of spherical fiducials with an increasing range of sizes (0.5–10 μm diameter). Larger particles (>5 μm diameter) diffract more and give rise to more pronounced diffraction rings. which facilitates accurate calculations. However, larger particles may require a larger field of view and fitting the profile slows down the acquisition and/or calculations, particularly for multi-fiducial tracking.

**Microscopy**. soSPIM module[4] was mounted on a conventional inverted microscope (Nikon Ti-E). Illumination lasers with wavelengths of 405 nm (200 mW; Vortran), 488 nm (200 mW; Coherent), 561 nm (200 mW; Vortran) and 640 nm (150 mW; Coherent) were passed through clean-up filters, combined via dichroic beam mirrors and coupled into optical fiber (Thorlabs, P3- 405BPM-FC-2). The lasers were collimated and aligned into a beam–steering system, which consists of a x1.5 beam expansion, an x and y-axis galvanometric mirror (Pangolin SCANMAX 506) and a tunable lens (EL-30-10, Optotune). These were conjugated to the back focal plane of the objective (1.27 NA, x60 CFI Nikon) so that the excitation beam was always emitted parallel to the objective optical axis[4]. The sample was mounted on a nano-positioning stage (Mad City Labs, LP50-200) atop a linear translation stage (ASI, LS50). The fluorescence was captured by the same objective and imaged using a CMOS camera (Hamamatsu Orca Flash 4). The transmission profile was recorded by illuminating fiducials within the wells with a 440 nm LED (CoolLED), separated from the fluorescence emission via a dichroic (Semrock Di03-R473-t3), filtered (Semrock, FF01-440/40-25) and focused ($f = 300$ mm) onto a CMOS camera (Allied Vision, Manta).

TIRF microscopy was achieved by displacing the laser, focused on the back focal plane of the objective, towards the periphery of the objective[10]. Briefly, laser lines of 405 nm (Vortran, Stradus 405–100), 488 nm (Vortran, Stradus 488–150) and 637 nm (Vortran, Stradus 637–180) were filtered, expanded ×10 and focused onto the back aperture of a 100 × 1.49 NA TIRF objective (Nikon, CFI Apochromat) using an achromatic lens ($f = 200$ mm), via a dichroic beamsplitter (Chroma, ZT488/640rpc). Displacement of the laser towards the edge of the objective was performed by moving the focusing lens with a mirror assembled on a translation stage (M-423-MIC; Newport). The sample was mounted on a nano-positioning stage (Mad City Labs, LP50-200) atop a micro-positioning stage (Mad City Labs, MicroStage). The fluorescence was captured by the same objective and focused ($f = 400$ mm) onto an EMCCD camera (Andor, iXon 897). For astigmatism based axial localization, a weak cylindrical lens ($f = 500$ mm) was placed before the EMCCD. The fiducial structures were illuminated with an infrared LED (Mightex LCS-0850-02-22), filtered (Semrock, FF01-842/56-25) and focused ($f = 200$ mm) onto a CMOS camera (Allied Vision, Manta).

Highly inclined and laminated optical sheet (HILO) microscopy[6] was performed using the same optical setup used for TIRF microscopy, as detailed above. Laser beams are focused on the back aperture of the objective which is translated towards the periphery of the objective. HILO creates a light sheet, providing optical sectioning capabilities superior to widefield imaging at depths above the coverslip.

All three acquisition methods used a single objective for the delivery of the laser excitation to the sample and subsequent acquisition of the fluorescence emitted.

An environmental control box was installed to control temperature fluctuations of the microscope. A series of Peltiers (TEC3-6; Thorlabs) with temperature transducers (AD590; Thorlabs) were regulated individually (TED200C; Thorlabs). Temperature and humidity were registered independently from the temperature control units (TSP01; Thorlabs). The temperature and humidity have a standard deviation of 0.02 °C and 0.88%, respectively. Before imaging, we allowed the samples to acclimatize for ~15 min.

**Direct STORM imaging**. All STORM imaging was performed in a closed chamber with buffer containing reducing and oxygen-scavenging compounds. The stained cells were imaged in PBS with the addition of 50 mM mercaptoethylamine, 5% glucose (wt/vol) and oxygen-scavenging enzymes (0.5 mg ml$^{-1}$ glucose oxidase (Sigma-Aldrich), and 40 mg ml$^{-1}$ catalase (Roche Applied Science)[33]. The ensemble fluorescence was converted into the desired density of single molecules by using the 640 nm laser at high power. For TIRF imaging, reduction was performed by briefly placing the setup into HILO and performing a z-stack with high laser power. For soSPIM imaging, we increased the power delivered via the light sheet and performed a z-stack prior to refocusing on the plane of interest using the printed fiducials as references. Prior to reducing dyes into a dark state, a standard TIRF or soSPIM light-sheet image was acquired. For image acquisition the laser power was reduced, and a UV laser (405 nm) turned on to aid switching. Approximately 10,000–20,000 images were recorded and analyzed.

**Image processing and single-molecule localization**. Fluorescent images were recorded and saved in a 16-bit TIFF format. To localize molecules, the point spread function of an individual molecule within each frame was fitted to a 2D or 3D elliptical Gaussian profile. STORM images were analyzed with the Picasso software[34] or ImageJ (NIH) with the ThunderSTORM plugin[35]. Single-particle tracking was performed using the TrackMate plugin[36] in ImageJ (NIH).

**Reporting summary**. Further information on research design is available in the Nature Research Reporting Summary linked to this article.

## Data availability

CAD designs, basic explanations on geometries and examples of individual STLs for the nano-fabricated fiducials can be found on GitHub (https://github.com/spcoelho/Direct-Laser-Writing-CAD-and-STL). Large SMLM raw data files are available from the corresponding author on reasonable request. Source data are provided with this paper.

## Code availability

The focus-locking software, including executables, can be found on GitHub (https://github.com/spcoelho/Active-Stabilization). We also include a CPU based ImageJ plugin for focus locking in on GitHub (https://github.com/spcoelho/Active-Stabilization) and in the Supplementary Software.

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

## Acknowledgements

We thank the support from the Australia Research Council (CE140100011 to K.G., FL150100060 and CE140100036 to J.J.G.) and National Health and Medical Research Council of Australia (APP1183588 to S.C. and K.G. APP1059278 to K.G. and APP1196648 to J.J.G.). We thank Doug Mair at the Australian National Fabrication Facility and Simon Thiele from Printoptics for their assistance in Nanoscribe. We also thank Virgile Viasnoff and Saburnisha Raffi (NUS/CNRS Singapore) for the soSPIM chambers. We also thank Jean-Baptiste Sibarita, Remi Galland (University Bordeaux), Varun Sreenivasan and Matthew Graus (UNSW) for the help in soSPIM imaging. We would like to thank Mara Catarino and Jesse Goyette for assistance with the paper preparation.

## Author contributions

S.C., J.B., and K.G. conceived of the project. S.C., J.G., and K.G. wrote the paper. S.C., J.B. and J.W. performed the experimental work and analyzed the experimental data. J.W. developed and validated the ImageJ plugin.

## Competing interests

The authors declare no competing interests.
