## [Peer Review File · Nature Communications]

Direct laser writing for subnanometer focusing and single-molecule imagingREVIEWER COMMENTS

Reviewer #1 (Remarks to the Author):

In this manuscript Simao Coelho et al propose to use prefabricated 3D structures on coverglass for subnanometer 3D focus control and single-molecule imaging. The authors claim that real-time isotropic subnanometer 3D focusing (< 0.8 nm) over tens of micrometers could be achieved from a combination of prefabricated 3D fiducial markers, automatic interference pattern recognition and nano-positioning stage. Although the use of two-photon direct laser writing method for nanofabricating three-dimensional structures on the surface of coverglass is not new, the proposed method of using these structures for subnanometer 3D focusing seems promising. The authors claim that this method is powerful for 3D imaging deep inside cells, but the data supporting this claim are not sufficient in the current manuscript. Results and discussions on the advantages and limitations of this 3D focus control method should be significantly strengthened, especially for single molecule imaging deep inside cells, for example, at depths > 5 μm from the coverslide (as mentioned in the introduction). The manuscript may be publishable in Nature Communications after a major revision.

Other concerns or comments:

1. An illustration showing how to make 3D fiducial markers on the coverglass surface and how to use 3D fiducial markers for focus determination and stabilization would be helpful. And, it is better to compare experimentally the performance of the proposed method with other reported focus locking methods.

2. Fig 1. What is sample stability and how to measure it? How to use the structures in Fig 1e-1f for 3D single molecule imaging? The challenge is not on how to make these complicated structures using a commercial device, but on how to use them in 3D single molecule imaging.

3. Fig 2. It would be interesting to quantify the transmitted diffraction profiles of the fiducial inside the box in different z positions. Showing data deep inside cells (> 5 μm) using biological samples would be more convincing to verify the power of the prefabricated nanostructures in 3D focus locking. Besides, the super-resolution image in Fig 2c is small and blurred. A bigger image with higher resolution is needed.

4. Fig 3. How to use the structures in Fig 3a in STORM imaging at different depths?

5. Methods – 3D focus locking. What are the maker and the model of the nano-positioning stage?

6. Fig S1. How to calculate the lateral stability and how to verify that these values are real? And, what is the difference between sample stability in Fig 1d and Fig S1? Are they from the same data set? The values seem to be different.

7. Fig S2. How does the height of the spherical cap affect sample stability? A detailed description could help readers understand the principle.

8. Fig S3. Any super-resolved image of microtubules deep inside cells? In Fig 3, biological structures from 8 μm , 12 μm and 18 μm depth were shown. Could the authors perform similar experiments using microtubule samples?

9. Fig S4. A calibration curve showing the precision and repeatability of axial offset estimation would be necessary. And, the routines of automatic surface detection and 3D focus lock are basically the same. However, the accuracy of automatic surface detection (35 nm, Fig S4) is 40 times larger than the accuracy of 3D focus lock (< 0.8 nm, Fig S1). Why?

10. Fig S5. Could the authors demonstrate how to use this extra-long fiducial structure in 3D single molecule imaging?

11. Supplementary Table 1. The authors said "The residual drift shown corresponds to out-of-loop reference fiducial acquired simultaneously." This residual drift does not seem to be able to represent stability. So, it would be necessary to discuss how to measure this stability and how to verify these numbers. And, the isotropic focus locking was evaluated on non-biological samples and through localization precision. It would be necessary to provide more evaluation using real biological samples with known structures in different depths, along with a calibration curve over a large depth.

12. The manuscript used direct laser writing to create various 3D optical structures and use transmitted light in brightfield to record the diffraction patterns from these structures. What is the measurement precision of this brightfield-based position estimation method? And, how to obtain quality control on the size of these nanostructures?

13. In the discussion section, the authors mentioned the production of complex three-dimensional structures that can span across millimeter ranges and claimed a potential application in fluorescence optical microscopy. Can the authors show some experimental data on this exciting direction, especially in single molecule fluorescence imaging? Imaging deep inside biological samples is not only limited by the drift stabilization, but also by signal collection efficiency and optical system aberrations.

Reviewer #2 (Remarks to the Author):

The authors present a very interesting approach to sub-nm precision focusing for single-particle tracking and live-cell imaging. The novelty is not in the tracking itself, as this is pretty standard, but in the cleverly designed fiducials, which first of all enable the positioning with highest accuracy due to reproducible shape of the fiducials. The reproducible shape for the first time allows the reliable use of look-up tables for the position reading of the sample. While the particle itself are imaged via a different technique, following the sample movement is highly important to disentangle sample movement from particle movement.

The manuscript is written in a clear and concise style and most of the required information is given to substantiate all the claims made by the authors.

Few points remain open and should be clarified prior to publication:

1. The software used to evaluate the position of the sample: is it commercially available or is it written in the lab? If the latter it would be great to get some more details about how the software identifies the position of the fiducials laterally and axially. In any case, it would be important to know how the look-up tables are trained and how long this training takes.

2. What are the most ideal fiducial shapes and sizes compared to the objective lenses used. As far as I understand, the contrast of the interference patterns is important for the evaluation algorithm to clearly identify the position of the sample. This will for sure change with the numerical aperture of the objective lens used for evaluation. Are there any desing-rules the author could provide or is it trial-and-error for each new objective?

With these two minor comments clarified the manuscript would be ready for publication in nature communications.

REVIEWER COMMENTS

Reviewer #1 (Remarks to the Author):

In this manuscript Simao Coelho et al propose to use prefabricated 3D structures on coverglass for subnanometer 3D focus control and single-molecule imaging. The authors claim that real-time isotropic subnanometer 3D focusing (< 0.8 nm) over tens of micrometers could be achieved from a combination of prefabricated 3D fiducial markers, automatic interference pattern recognition and nano-positioning stage. Although the use of two-photon direct laser writing method for nanofabricating three-dimensional structures on the surface of coverglass is not new, the proposed method of using these structures for subnanometer 3D focusing seems promising. The authors claim that this method is powerful for 3D imaging deep inside cells, but the data supporting this claim are not sufficient in the current manuscript. Results and discussions on the advantages and limitations of this 3D focus control method should be significantly strengthened, especially for single molecule imaging deep inside cells, for example, at depths > 5 μm from the coverslide (as mentioned in the introduction). The manuscript may be publishable in Nature Communications after a major revision.

We want to thank the reviewer for the positive assessment of our manuscript. We have significantly modified the text of manuscript, performed comparisons to previous approaches and have included additional experiments to further validate our 3D focusing approach. Briefly:

1. New Fig 1, Fig. 4 and Supplementary Fig 1 include illustrations showing the fiducial structure fabrication, how these are used to focus lock and how single objective selective plane illumination microscopy was performed
2. We compared our approach by performing experimental comparisons to commercial focus locks (Nikon Perfect Focus), active stabilization using image correlation (Supplementary Fig. 6). We validated our method by using fluorescent fiducial markers deposited on top of a cell layer (Supplementary Fig. 7).
3. Descriptions detailing how 3D focus locking is performed, controls and validation
4. Expanded Discussion section including advantages, limitations and future directions.

Other concerns or comments:

1. An illustration showing how to make 3D fiducial markers on the coverglass surface and how to use 3D fiducial markers for focus determination and stabilization would be helpful. And, it is better to compare experimentally the performance of the proposed method with other reported focus locking methods.

We have included an illustration on the a) fiducial structure fabrication (Fig. 1a and Supplementary Fig 1) and b) how these are used (Fig 1b and Fig. 4a). Briefly,

- a) Light from a two-photon laser is used to trigger a chemical reaction that causes polymerization of a photosensitive material. The nonlinear excitation occurs at the focal

point leaving other regions unaffected by the laser light. This phenomenon reduces solidification resolution to below the diffraction limit of the applied light. By selectively solidifying the photo-resin, 3D geometries are produced in a layer-by-layer approach. At the end of the solidification process, photoresist which is not solidified, is washed away, leaving the desired structures behind. This additional explanation is included in the methods section under ‘Direct laser writing’. On line 282:

‘Light from a two-photon laser two-photon laser light is used to trigger a chemical reaction that causes polymerization of a photosensitive material. The nonlinear excitation occurs at the focal point leaving other regions unaffected by the laser light. This phenomenon reduces solidification resolution to below the diffraction limit of the applied light. By selectively solidifying the photo resin, 3D geometries are produced layer-by-layer’

- b) The structures are illuminated using brightfield. This creates pronounced diffraction patterns which are detected using a separate imaging channel. Fig. 1b and Fig. 4a illustrates the projection of the diffraction pattern and how focusing is performed.

We now include an experimental characterization of 1) a commercially available focus lock (Nikon – Perfect Focus System); 2) brightfield using image correlation as described in McGorty et al. 3) Using our focusing method we deposited fluorescent fiducials above a cell layer, at a depth of 5 μm for ~ 3 hours and monitored drift as a function of time.

Addressing each of these points in turn,

- 1) The Nikon Perfect focus system (N-PFS)

The N-PFS is based on the reflection of an infrared LED off the surface of a coverglass. Sample movement, in the axial direction, is recorded by monitoring the deviations of the LED. This allows for 22 nm of z- stability, as shown in Supplementary Fig. 6a-b (found below).

We also note that, unfortunately, using the reflection of the coverglass is limited to the axial direction. Movement in the x-y direction is not corrected. Furthermore, if there happens to be a tilt in the coverglass or reflections from within the sample (i.e. soSPIM mirrors) the axial stabilization via this approach can produce additional errors.

- 2) 3D stabilization comparison with active microscope stabilization using image correlation.

McGorty et al. 2013 have shown an elegant approach where the contrast of the cell is used to correct for an estimate of the movement sensed by the sample. The stage is then repositioned, therefore focus locking. However, as described in McGorty et al, cells provide limited contrast and the method is not compatible with live cell imaging.

In Supplementary Fig. 6c-d (found below), we have replicated the method described in McGorty et al and obtain a stabilization of 13 nm and 19 nm in the lateral and axial directions, respectively. These are comparable to previously reported values. As described in McGorty et al,

this is due to the limited resolution provided by the low contrast of cells in brightfield images. In addition, the cell-based image correlation method cannot work in low contrast environments (e.g. DNA-origami) and is not compatible with many single-molecule techniques (i.e. live PALM or single particle tracking). We present a substantial improvement (~ 20 fold) in stabilization and without these limitations.

Supplementary Figure 6: 3D focus locking experimental comparison. **a-b**, Nikon Perfect Focus System. **a**, Lateral (x-direction) and **b**, axial stability of Nikon Perfect Focus determined by monitoring a fiducial in parallel. The standard deviations are 124 nm and 22 nm, respectively. **c-d**, the cell-based image correlation correction. Focus locking was performed using the contrast of cellular structures in three dimensions via image correlation. Drift was monitored in parallel using an out-of-loop fiducial (grey dots, 1/1000 point shown). The standard deviations in lateral **c** and axial **d** directions are 13 nm and 19 nm, respectively. This is comparable to previously published data (McGorty et al, 2013).

3) Fluorescent fiducials.

We deposited fluorescent fiducials on top of a cellular layer and, using our fabricated fiducials, we focused locked. Over ~ 3 hours, we recorded the position of the fiducial and found < 2 nm standard deviation in the position recorded. The accuracy of ~ 2 nm is consistent with the photon limited localization precision registered from the fiducial. This is shown in Supplementary Figure 7 (added below).

Supplementary Figure 7: Experimental validation of the 3D focus locking. Fluorescent beads were deposited on top of cells. With the 3D focus lock engaged we monitored the position of the fiducial as a function of time. **a**, Widefield image showing a fluorescent bead above a cell at a depth of 5 μm . Scale bar = 3 μm . **b**, Brightfield image of the cell. **c** and **d**, Gaussian fitting of the fiducial indicates a standard deviation of 1.95 nm in the x direction and 1.75 nm in the y direction after ~ 3 hours. This is consistent with the photon limited localization precision registered, thus demonstrating that the fluorescence remained focused and stable.

The three points above were added to manuscript. On line 153:

‘To compare the performance our of approach to commonly used focus locking mechanisms, we performed experimental evaluations of 1) a commercially available focus lock (Nikon Perfect Focus System) and 2) image correlation correction using the contrast from cellular structures (Supplementary Figure 6). We found our method improves on commercially focus locks and image correlation by ~ 20 -fold. To validate our approach, we deposited fluorescent fiducials on top on cells at a depth of 5 μm (Supplementary Figure 7). Over ~ 3 hours, we registered < 2 nm xy variation in position. This is consistent with the photon limited localization precision recorded, thus demonstrating that the fluorescence remained focused and stable.’

2. Fig 1. What is sample stability and how to measure it? How to use the structures in Fig 1e-1f for 3D single molecule imaging? The challenge is not on how to make these complicated structures using a commercial device, but on how to use them in 3D single molecule imaging.

The stability of the sample is determined by the standard deviation of the position of a fiducial, recorded as a function of time. The first structure selected is considered to be the 'lock'. The 'lock' is the fiducial that the software is actively correcting for. Therefore, movement registered by the 'lock fiducial' will result in the stage repositioning the sample. Here, the error is the difference between the current and original position (x/y/z) at the start of the experiment. To independently register the sample position, a second fiducial can be chosen. This second structure is referred to as the 'reference'. The 'reference' is an out-of-loop fiducial whose deviations are calculated independently from the 'lock'. In our experience, we find the performance of both approaches to be comparable. The average drift over a period of ten frames is subtracted from the stage position at a rate of ~20 Hz. Applying an equal but opposite amplitude on the sample stage suppress drift/sample movement and allows for accurate focus-locking. Sample fluorescence is acquired independently from the focus-locking routine allowing for focus-locking to be performed in parallel.

We have modified the text to highlight this. On line 85 and in the methods section:

'The stability of the sample is determined as the standard deviation of the position of the fiducial recorded as a function of time. The sample stability is recorded independently using a second structure as a reference. The reference structure is an out-of-loop fiducial whose deviations are calculated independently from the focus locking routine. Focus locking can also be monitored by recording the difference between the current and original position (x/y/z) at the start of the experiment.'

The fiducial arrays, as shown in now Fig. 2e-f, are a demonstration of how an array of different fiducials can be organized in a pre-determined design. Also, having fiducial arrays arranged in a known way across the slide facilitates software automation as these are arranged in a designated geometry. Each individual fiducial can be used to focus lock using their brightfield profile (Supplementary Videos).

To make this clear, we have modified the text and included Figure 1b. On line 97:

'By producing fiducial arrays with different geometries, dimensions and distributions, we created prefabricated coverglass capable of focus locking, via their brightfield profile, across a large range of imaging volumes (Fig. 1e and f and Supplementary Video 2 and 3).'

3. Fig 2. It would be interesting to quantify the transmitted diffraction profiles of the fiducial inside the box in different z positions. Showing data deep inside cells (> 5 μm) using biological samples would be more convincing to verify the power of the prefabricated nanostructures in 3D focus locking. Besides, the super-resolution image in Fig 2c is small and blurred. A bigger image with higher resolution is needed.

We have modified Fig 3c and highlighted one of the panels.

In our new Supplementary Video 4, we show the profiles of 9 fiducials with pedestals with varying heights and diameters all inside protection boxes.

To validate our focus-locking routine beyond the coverslide, we deposited fluorescent fiducials atop of cell layer. Fiducials were used as these are photostable and allow for long acquisitions. We imaged fluorescent fiducials at a depth of 5 μm for ~ 3 hours and found that our focusing surpasses the photon limited localization precision of 2 nm, thus validating the focus locking. Also, in Figure 4, we already show cells at different depths therefore demonstrating that the diffraction profiles are of sufficient quality to image at depths of 18 μm

4. Fig 3. How to use the structures in Fig 3a in STORM imaging at different depths?

We have included an illustration in, now, Fig. 4 which demonstrates how 1) the structures are used and 2) fluorescent imaging is performed. Briefly, laser light is reflected off a 45° mirror selectively illuminating a single fluorescent plane within the sample. Focus locking onto the axial planes is done by using the diffraction pattern of a 3D fiducial.

5. Methods – 3D focus locking. What are the maker and the model of the nano-positioning stage?

The nanopositioning piezo stage is Mad City Labs (Nano-LP) with 50 μm custom x/y axis range (0.1 nm step size) controlled using a USB 20-bit upgrade. A description of the stage is included in the methods section under *Microscopy*.

6. Fig S1. How to calculate the lateral stability and how to verify that these values are real? And, what is the difference between sample stability in Fig 1d and Fig S1? Are they from the same data set? The values seem to be different.

As described in point 2, the 3D stability is the standard deviation of the position determined using a second out-of-loop structure acquired in parallel from the focus-locking routine.

In Supplementary Figure 2, we show graphs which describe the x/y axis (lateral) and z-axis (axial) as a function of time. Here, each z-plane shown was acquired over >1 day. This demonstrates the long-term stability of the focus-locking.

To avoid confusion, we have modified the description of the figure. Now Supplementary Figure 2 clarifies that 1) it shows the long-term stability of the focus-locking and 2) added the time frame on which the experiment was acquired.

Fig. 2d (previously Fig. 1d) is the result of multiple experiments. Fig. S2 is comparable to one of the experiments used in Fig. 2d, however over a different time period. The standard deviations registered between Fig. 2d and Fig. S2 are comparable but taken on different days, hence the slightly different values (± 0.2 nm).

7. Fig S2. How does the height of the spherical cap affect sample stability? A detailed description could help readers understand the principle.

The change in spherical cap gives rise to a change in contrast, consequently sample stability is increased with improved signal-to-noise. Our understanding is that the signal variation stems from the range of incident/exit angles of the brightfield illumination at glass-polymer interface. We have included this in the legend of the updated figure.

8. Fig S3. Any super-resolved image of microtubules deep inside cells? In Fig 3, biological structures from 8 μm , 12 μm and 18 μm depth were shown. Could the authors perform similar experiments using microtubule samples?

This experiment is feasible but requires a 3D culture method in which cells spread and have a well-organized microtubule network. The microtubule network very much depends on the 3D matrix (see Mierke et al 2018 and Raab et al 2017). Analyzing and comparing different matrices we feel is beyond the scope of this work and we believe it would not substantially improve the paper as a result of the already demonstrated feasibility.

9. Fig S4. A calibration curve showing the precision and repeatability of axial offset estimation would be necessary. And, the routines of automatic surface detection and 3D focus lock are basically the same. However, the accuracy of automatic surface detection (35 nm, Fig S4) is 40 times larger than the accuracy of 3D focus lock (<0.8 nm, Fig S1). Why?

The routine for finding the surface and stabilizing are similar, however are different routines. Stabilization holds the sample in place and is how well the system is focus-locked (in 3D). Here, the performance is compared to an out-of-loop fiducial (as explained in the second point).

Obtaining the surface automatically is performed by assuming a diffraction profile and obtaining successive iterations to the estimated position of the surface. Errors from the estimated position of the surface and the surface itself can arise from a variety of factors such as surface flatness. The precision of the routine in finding the surface is 35 nm, as demonstrated in Supplementary Fig. 5c (previously, Fig. S4). For the same fiducial, the repeatability is comparable to the focus locking routine. These issues can be circumvented by manually (re)focusing onto the fluorescence from the sample prior to imaging.

We have rewritten the manuscript to address this. In methods section, under ‘Unsupervised data acquisition’:

‘The accuracy of the automatic surface detection routine is 35 nm (Supplementary Figure 5) and was determined by recording the height difference between our autonomous focusing method and the position with maximum signal of individual molecules deposited onto the coverslide. Errors from the estimated position of the surface and the surface itself can arise from a variety of factors such as surface flatness. In practice, this error can be reduced by manually

(re)focusing onto the fluorescence from the sample prior to imaging. The focus-locking routine, which maintains the sample stationary, is activated after the fiducial is found on the surface and the performance is not influenced by the variations of the automatic surface detection routine.'

10. Fig S5. Could the authors demonstrate how to use this extra-long fiducial structure in 3D single molecule imaging?

Unfortunately, piezo stages with subnanometer step sizes have a limitation on the range. Currently, nanopositioner manufactures balance travel range and step accuracy. Consequently, travel range is sacrificed to allow for Ångstrom step size precision. To fully capture extra-long fiducials, this would currently require a stage with reduced accuracy in exchange for a long travel range. We are confident that future developments in nanopositioner and fluorescent imaging technologies will alleviate this limitation.

11. Supplementary Table 1. The authors said “The residual drift shown corresponds to out-of-loop reference fiducial acquired simultaneously.” This residual drift does not seem to be able to represent stability. So, it would be necessary to discuss how to measure this stability and how to verify these numbers. And, the isotropic focus locking was evaluated on non-biological samples and through localization precision. It would be necessary to provide more evaluation using real biological samples with known structures in different depths, along with a calibration curve over a large depth.

Here, residual drift and sample stability were used interchangeably. As detailed in Point #2, we determine the focusing stability by using a second structure acquired in parallel. The second structure is an ‘out-of-loop’ fiducial that provides information on the sample’s position. Also, using fluorescent beads (as shown in Supplementary Fig. 7), we demonstrate focus-locking independently from the focus locking routine. Unfortunately, evaluating long term stability on a biological sample is not viable as these have poorly defined shapes and bleach over time, thus do not reflect stability accurately.

In order to avoid misunderstanding over terminology, in Supplementary Table 1 we have replaced ‘residual drift’ for ‘standard deviation’.

12. The manuscript used direct laser writing to create various 3D optical structures and use transmitted light in brightfield to record the diffraction patterns from these structures. What is the measurement precision of this brightfield-based position estimation method? And, how to obtain quality control on the size of these nanostructures?

The accuracy to determine the structures position via brightfield is in the Ångstrom regime, as shown by Huhle et al (Nat Commun 6, 5885, 2015).

The device used to produce the structures (Nanoscribe) has an accuracy between 160-200 nm, which is the ‘solidification volume’ of a two-photon laser. This is the 3D accuracy routinely achieved using the Nanoscribe. Careful scanning allows for structures to be produced with a surface roughness of 13.0 nm (root-mean squared), as shown in Gissibl et al (Nature Photon 10,

554–560, 2016). This is also evident in the SEM images in Fig. 2. Decreasing the layer-by-layer distance also improves the accuracy of the structure. Previous studies (e.g. Juodkazis et al Nanotechnology, 2005) have shown that by manipulating materials and fabrication techniques, finer structures with features of ~ 30 nm have been successfully produced.

We have changed the text to highlight this. On line 68:

'3D samples have an accuracy between 160-200 nm, which is routinely achieved using the Nanoscribe. Careful scanning allows for structures to be produced with a surface roughness of 13.0 nm [14]. Previous studies have shown that by manipulating materials and fabrication techniques, finer structures with features of ~ 30 nm can be produced successfully'

13. In the discussion section, the authors mentioned the production of complex three-dimensional structures that can span across millimeter ranges and claimed a potential application in fluorescence optical microscopy. Can the authors show some experimental data on this exciting direction, especially in single molecule fluorescence imaging? Imaging deep inside biological samples is not only limited by the drift stabilization, but also by signal collection efficiency and optical system aberrations.

The reviewer is correct in pointing out that the localization precision of single molecule techniques can be influenced by other factors such as optical aberrations. Optical techniques with high collection efficiency and penetration, such as lattice light sheet with adaptive optics (Liu et al, Science 2018), will be important to push the field towards high resolution single-molecule imaging. Other approaches such as high-photon count DNA-PAINT in tissue (Wang et al, Nano Lett, 2017) may also prove important for accuracies <10 nm in 3D imaging. These approaches can benefit from 3D focus-locking with high accuracies. Unfortunately, we don't have the equipment available to perform millimeter range single-molecule imaging.

As a result, we have changed the text to highlight this. On line 204

'Currently, achieving highly accurate structures using direct laser writing can be a slow, expensive process as fabrication is performed layer-by-layer using two-photon lithography. We expect that advancements in fabrication procedures, including parallelization [25] and new photopolymers can make this more accessible to wider scientific community and even mass-producible [26]. It is also worth noting that incorporating high accuracy focus locking can prove beneficial to improving optical techniques with high collection efficiency and penetration, such as lattice light sheet with adaptive optics [27] and/or high photon count DNA-PAINT approaches[28].'

Reviewer #2 (Remarks to the Author):

The authors present a very interesting approach to sub-nm precision focusing for single-particle tracking and live-cell imaging. The novelty is not in the tracking itself, as this is pretty standard, but in the cleverly designed fiducials, which first of all enable the positioning with highest accuracy due to reproducible shape of the fiducials. The reproducible shape for the first time allows the reliable use of look-up tables for the position reading of the sample. While the particle itself are imaged via a different technique, following the sample movement is highly important to disentangle sample movement from particle movement.

The manuscript is written in a clear and concise style and most of the required information is given to substantiate all the claims made by the authors.

Few points remain open and should be clarified prior to publication:

1. The software used to evaluate the position of the sample: is it commercially available or is it written in the lab? If the latter it would be great to get some more details about how the software identifies the position of the fiducials laterally and axially. In any case, it would be important to know how the look-up tables are trained and how long this training takes.

We would like to thank the reviewer for the positive review of our manuscript.

The software was written in the lab. Different fiducial structures create different diffraction patterns upon brightfield illumination, as demonstrated in the supplementary videos. Consequently, look-up tables (LUT) are required for different structures. For spherical fiducials, the simplest case, there is a radial symmetry in the diffraction pattern characterized by pronounced diffraction rings. For a given fiducial size, the diameter of the rings and the spacing between these indicate the fiducial's position. If the sample moves axially, the widths of the ring's increase/decrease. Movement in the axial direction is quantified by comparing the diffraction pattern to an axial LUT. If the sample moves in the lateral directions, the rings are displaced in x/y. The lateral positions are determined by correlating the linear intensity profiles with their mirror profiles (described in Huhle et al). This allows us to determine the axial and lateral position of an individual fiducial.

For more complex structures, such as pedestals, the focus-locking procedure is comparable, however includes an additional term to compensate for the axial offset. For the remaining non-symmetrical structures, the diffraction pattern will have a LUT which matches their geometries. For example, cube structures will have a square diffraction pattern.

Fiducials are first identified by 1) their spacing, 2) predicated geometry, 3) distribution throughout the slide (in relation to each other) and 4) their xy centre is initially estimated using cross-correlation and z via the estimated look-up table. After a fiducial is identified we determine their 3D position by via their individual look-up table.

We have incorporated these comments into the manuscript. Specifically,
On line 356:

'Initially, the xy position of a fiducial is estimated, within ~ 1 pixel, by using cross-correlation. Fiducials are then identified by their spacing, predicated geometry and distribution throughout the slide. After a fiducial is identified their 3D position is determined via their individual look-up table.'

The 'searching' look-up tables are the result of sequential averaging. These are then used to estimate the fiducial's position. An experimental look-up table is then acquired, saved and added to estimate the next fiducial. In practice, the learning phase is based on collecting multiple fiducials with the same characteristics, acquired under the same conditions. Naturally, the more fiducials used with the same properties makes the estimation more reliable. Typically, we average 10-30 fiducials.

We have incorporated these comments into the manuscript. Specifically, On line 353:

'Prior to determining a new fiducial's position, we typically use an average of 10-30 diffraction profiles with the same geometries.'

2. What are the most ideal fiducial shapes and sizes compared to the objective lenses used. As far as I understand, the contrast of the interference patterns is important for the evaluation algorithm to clearly identify the position of the sample. This will for sure change with the numerical aperture of the objective lens used for evaluation. Are there any design-rules the author could provide or is it trial-and-error for each new objective?

Here, the reviewer makes a great comment. The contrast is clearly important for the approach to perform well. Different objectives give rise to a different contrast due to their NA/magnification/refractive index media. In this work, we have used only objectives with high numerical aperture (1.49 and 1.27 NA) which were previously inspected by the manufacture using wavefront aberration technologies to ensure the lowest possible asymmetric aberration and superior optical performance required for super-resolution imaging.

When using a new objective, we first evaluate it using an array of spherical fiducials with multiple sizes. This allows us to evaluate the focus-locking performance as a function of fiducial size. Larger individual fiducials allow for more pronounced diffraction patterns, however if these are too large it can slow down the focus-locking and reduce performance.

We also use the multi-size array to adjust other experimental parameters such as collar correction of the objective (to minimize spherical aberration) and evaluate additional corrections (e.g., 0.7 for water-oil). Although the impact of the objective (and resulting contrast) can be simulated in software, we find that first calibrating using an array of fiducials with multiple shapes and sizes is helpful to build look-up tables and get good performance.

Increasing the contrast can also be achieved using high refractive index photoresist (e.g., Nanoscribe IP-n162, with a refractive index of 1.62). Nanofabrication and photopolymer optimization are an area of intense research and we expect advances to follow.

With regards to specific shapes/sizes, we don't have any particular design rules. Polystyrene beads have been extensively used to monitor 3D movement of DNA-tethered particles. Dimensions for individual fiducials are typically in the 2-6 μm range.

We have now included these points in the revised manuscript. Specifically:

In the methods section, on line 397:

'To ensure high contrast in brightfield, we selected high N.A. objectives. These were previously inspected by the manufacture using wavefront aberration technologies to ensure the lowest possible asymmetric aberration and superior optical performance required for super-resolution imaging.'

The diameter of the fiducial can influence performance. To determine the best fiducial dimension for a given objective/magnification, we empirically evaluate each imaging objective using an array of spherical fiducials with an increasing range of sizes (0.5 – 10 μm diameter). Larger particles (>5 μm diameter) diffract more and give rise to more pronounced diffraction rings, which facilitates accurate calculations. However, larger particles may require a larger field of view and fitting the profile slows down the acquisition and/or calculations, particularly for multi-fiducial tracking.'

REVIEWER COMMENTS

Reviewer #1 (Remarks to the Author):

Compared to the first version, the quality of the revised manuscript has been significantly improved, especially on 3D focus locking controls and validation. However, the performance of this new focus locking method at depths $> 5 \mu\text{m}$ from the coverslip (mentioned in the introduction section) are still not well quantified. The authors should perform and analyzed more imaging results from single particle tracking and/or 3D dSTORM at depths $> 5 \mu\text{m}$. The TIRF or HiLo microscopy results at depth $< 3 \mu\text{m}$, which are shown in Fig. 3, could be achieved by other (and more convenient) focus locking methods. The manuscript still requires further improvement before publishable in Nature Communications.

Other concerns or comments:

1. Line 154. Why did the authors use the name of soSPIM-dSTORM? If I understood it correctly, here the authors used only light-sheet illumination, not the whole technique in soSPIM (including both illumination and detection).
2. Line 159. What is "The 3D soSPIM single-molecule localization data"? In the Methods section, the authors described soSPIM module, but did not mention how to perform soSPIM-dSTORM experiments. HiLo microscopy was also not described.
3. Line 420. It is incorrect to say "two-photon laser light" or "two-photon laser". Saying NIR laser (780 nm), or similar, would be better.

Reviewer #2 (Remarks to the Author):

In this revised version the authors complied with all the requests made. The manuscript improved even further and especially all the details provided in the revised version about addressing the fiducials, using them with different microscope objectives and the quantitative evaluation of competing technologies really make this manuscript ready for publication. I am convinced that it will be a very well received paper by the community.

REVIEWER COMMENTS

Reviewer #1 (Remarks to the Author):

Compared to the first version, the quality of the revised manuscript has been significantly improved, especially on 3D focus locking controls and validation. However, the performance of this new focus locking method at depths $> 5 \mu\text{m}$ from the coverslip (mentioned in the introduction section) are still not well quantified. The authors should perform and analyzed more imaging results from single particle tracking and/or 3D dSTORM at depths $> 5 \mu\text{m}$. The TIRF or HiLo microscopy results at depth $< 3 \mu\text{m}$, which are shown in Fig. 3, could be achieved by other (and more convenient) focus locking methods. The manuscript still requires further improvement before publishable in Nature Communications.

We thank the reviewer for their review.

Here, there are two parts which need to be addressed individually. First, is the performance of the focus locking mechanism. Second, is the single-molecule imaging results/analysis at depths $> 5 \mu\text{m}$. Addressing each point in turn:

First, quantifying the performance of the focus locking mechanism: Throughout the manuscript, we have already demonstrated – at several instances – the performance of the focus locking method between $0 - 20 \mu\text{m}$. In Figure 2, supplementary figures 2 and 7 and supplementary table 1, we validate the performance of the focus locking mechanism. We do this by using out-of-loop structures as controls, and/or fluorescent fiducials as references. The out-of-loop fiducials demonstrate with Angstrom precision that the standard deviation of the sample is $< 1 \text{ nm}$ in 3D over $20 \mu\text{m}$, while the fluorescent fiducial taken at a depth of $5 \mu\text{m}$ shows an accuracy better than the photon limited localization precision recorded ($< 2 \text{ nm}$). In addition, we also compared our approach to previously published methods and show ~ 20 -fold improvement in all dimensions.

Collectively, we believe that these figures adequately satisfy the requirements of the referee as they 1) quantify the focus locking mechanism to have sub nanometer accuracy, 2) demonstrate the mechanism is reproducible, and 3) show the capability over a wide range of depths.

Second, is regarding single-molecule acquisitions $> 5 \mu\text{m}$ (e.g., dSTORM). For this we combined our focusing locking method with single-molecule imaging with increased depth imaging. We opted for single-objective selective-plane illumination microscopy (soSPIM), for 3 reasons: a) soSPIM has a penetration depth of tens of micrometers, b) the 45-degree geometry of the soSPIM micromirrors mean that movement during acquisition in the x-axis

can be misinterpreted for an axial offset and vice-versa and c) to demonstrate the feasibility of our method by incorporating it on an already established imaging technique. For a), a laser beam is reflected off a micromirror placed at 45-degrees. This creates a beam parallel to the aperture of the imaging objective. By using a rapid galvo-scanner a light sheet is created. 3D imaging is performed by laterally translating the laser across the micromirror. Focus-locking is performed in parallel via fiducials printed within the sample chambers adjacent to the micromirrors. Here, we use the full excitation/detection scheme for soSPIM and perform dSTORM acquisitions of CD45 tagged with Alexa-647. In Figure 4, we show light sheet soSPIM images and its respective dSTORM reconstruction acquired at depths between 8-18 μm . This is above the 5 μm range described in the introduction, thus showing that the focus locking mechanism works for single molecule acquisitions at depth. For b), soSPIM is performed by reflecting a laser off a 45-degree mirror – as described above. However, movement in the lateral direction leads to the reflected excitation light sheet to be offset to a new z-plane. Consequently, the focus is erroneously shifted via a movement in the x-direction. Also, an axial movement can be misinterpreted as an x-shift – as there is no clear point of reference. Our focusing method solves this issue. By having clearly defined 3D geometries which span over tens of micrometers, we can focus lock on the structures instead of using a standard focus locking mechanism such as the Nikon Perfect Focus System – shown in Supplementary figure 6. For c), we demonstrate our method is versatile and can be used in restricted environments – such as within the soSPIM micromirror chambers. As such, the method can be further employed in other sample geometries or acquisition techniques.

Taking all these points into account, we quantify the focus locking, demonstrate the improvement and performance and successfully show its viability for single-molecule imaging at depths beyond $> 5 \mu\text{m}$. We did consider doing further experiments to reemphasize these points but as a result of the severe COVID lockdown in Sydney (which promises to go on for many months), this was not possible as the laboratory for this work is closed.

Other concerns or comments:

1. Line 154. Why did the authors use the name of soSPIM-dSTORM? If I understood it correctly, here the authors used only light-sheet illumination, not the whole technique in soSPIM (including both illumination and detection).

We actually performed the entire soSPIM approach as previously described by Gallard et al 2015. This includes both the illumination and detection via a single objective. We suspect

the uncertainty by the referee arose from the additional light-source used for the focus-locking.

The soSPIM panels (Fig 4c), are the light sheet image plane acquired fluorescence prior to converting the Alexa-647 molecules into a dark state using a 640-nm laser. The soSPIM-dSTORM (Fig 4d) is the dSTORM acquisition of the panels above, hence the soSPIM-dSTORM nomenclature.

Briefly, the soSPIM encapsulates the light sheet approach used to illuminate the fluorescent molecules within the sample. As previously demonstrated by Gallard et al 2015, this light sheet approach allows for imaging in addition to performing dSTORM at depth. We replicated the previously described procedure and continued with the use of the already established terminology (i.e. soSPIM, soSPIM-dSTORM etc.).

To avoid potential misunderstandings, we have modified the text. Specifically:

On line 153: *'Fluorescence from each individual excitation light sheet plane was collected using the same high numerical aperture objective (Fig. 4a).'*

On line 155: *'We were able to nanoprint optical fiducials directly within individual wells of the imaging chambers (Fig. 4b) and performed light sheet imaging using standard soSPIM (Fig. 4c) and soSPIM-dSTORM over multiple axial sections'*

On line 425: *'All three acquisition methods used a single objective for the delivery of the laser excitation to the sample and subsequent acquisition of the fluorescence emitted.'*

On the legend for figure 4: *'Figure 4: 3D focus locking for super-resolution single-objective selective-plane illumination microscopy over cellular volumes. a, Laser light is reflected off a 45° mirror selectively illuminating a single plane within the sample. Fluorescence is captured via the same objective. Focus locking onto different axial planes is performed by using the 3D fiducial'*

2. Line 159. What is "The 3D soSPIM single-molecule localization data"? In the Methods section, the authors described soSPIM module, but did not mention how to perform soSPIM-dSTORM experiments. HiLo microscopy was also not described.

The dSTORM experiments, including soSPIM-dSTORM, was performed as previously described in Gallard et al, Nat. Methods 2015, based on the protocol from Rust et al, Nat.

Methods 2006. We now specify more clearly how these experiments were performed. On line 428,

'Direct STORM imaging. All STORM imaging was performed in a closed chamber with buffer containing reducing and oxygen-scavenging compounds. The stained cells were imaged in PBS with the addition of 50 mM mercaptoethylamine, 5% glucose (wt/vol) and oxygen scavenging enzymes (0.5 mg ml⁻¹ glucose oxidase (Sigma- Aldrich), and 40 mgml⁻¹ catalase (Roche Applied Science), as previously described[33]. The ensemble fluorescence was converted into the desired density of single molecules by using the 640 nm laser at high power. For TIRF imaging, reduction was performed by briefly placing the setup into HILO and performing a z-stack with high laser power. For soSPIM imaging, we increased the power delivered via the light sheet and performed a z-stack prior to refocusing on the plane of interest using the printed fiducials as references. Prior to reducing dyes into a dark state, a standard TIRF or soSPIM light sheet image was acquired. For image acquisition the laser power was reduced, and a UV laser (405 nm) turned on to aid switching. Approximately 10,000-20,000 images were recorded and analyzed.'

Also, we now include how the HILO microscopy was performed. On line 418,

'Highly inclined and laminated optical sheet (HILO) microscopy was performed as previously described[6]. HILO microscopy was performed using the same optical setup used for TIRF microscopy, as detailed above. Briefly, laser beams are focused on the back aperture of the objective which is translated towards the periphery of the objective. HILO creates a light sheet, providing optical sectioning capabilities superior to widefield imaging at depths above the coverslip.'

We also further clarify that the three acquisition approaches used a single objective for the delivery of the laser excitation to the sample and subsequent acquisition of the fluorescence emitted. On line 425,

'All three acquisition methods used a single objective for the delivery of the laser excitation to the sample and subsequent acquisition of the fluorescence emitted.'

3. Line 420. It is incorrect to say "two-photon laser light" or "two-photon laser". Saying NIR laser (780 nm), or similar, would be better.

We have amended the text, now line 450, to 'NIR femtosecond laser'.

Reviewer #2 (Remarks to the Author):

In this revised version the authors complied with all the requests made. The manuscript improved even further and especially all the details provided in the revised version about addressing the fiducials, using them with different microscope objectives and the quantitative evaluation of competing technologies really make this manuscript ready for publication. I am convinced that it will be a very well received paper by the community.

We thank the reviewer for their comments and help in improving the manuscript.

REVIEWERS' COMMENTS

Reviewer #1 (Remarks to the Author):

In this revision, the authors have addressed all of my concerns. This revised manuscript is now ready for publication.

Reviewer #1 (Remarks to the Author):

In this revision, the authors have addressed all of my concerns. This revised manuscript is now ready for publication.

We thank the reviewer for their comments, feedback and help to improve the manuscript.